# Ocular delivery of lipid nanoparticles-formulated mRNA encoding lanosterol synthase ameliorates cataract in rats

Ruiteng Song[1], Yongqi Lin[2], Min Zhang [3], Zhen Liu [1], Rui Zhang[1], Jun Zhao[3] & Bin Li [1,2] ✉

Cataract caused by crystallin aggregation is the leading cause of vision impairment and blindness globally. The only available treatment option so far is surgery. In this study, we leverage lipid nanoparticles (LNPs)-formulated mRNA encoding human lanosterol synthase (hLSS) to elevate lanosterol levels in the lens as a potential anti-cataract therapy. hLSS mRNA delivered with aromatized LNPs can be avidly taken up and translated into hLSS proteins in mammalian cells. mRNA formulations administered via intravitreal, sub-conjunctival, intracameral, or subretinal injection in rats display distinct kinetics and bio-distribution profiles, among which intracameral injection achieves sustained and selective protein expression in the lens. In comparison to clinically used LNPs, aromatized LNPs show more than seven-fold higher mRNA delivery potency in rats upon intracameral injection, without inducing significant ocular lesions. Furthermore, ocular delivery of hLSS mRNA-loaded formulations leads to elevated levels of hLSS proteins and lanosterol within the lens and a remarkable improvement in cataract symptoms in two rat models of cataract. Collectively, topical delivery of hLSS mRNA-LNPs to the eyes offers a potential strategy to reduce intracellular aggregation of crystallins and ameliorate cataract development.

The lens is a transparent, biconvex structure responsible for focusing light on the retina[1,2]. Cataract, also known as lens opacification, is a common age-related ocular disease characterized by a cloudy appearance and reduced transparency, representing the leading cause of vision impairment and blindness worldwide[1–3]. It is estimated that over 95 million people suffer from cataract globally[4]. While aging and oxidative stress are primary causes, additional risk factors such as genetic mutations, trauma, radiation, long-term corticosteroid use, and metabolic abnormalities can disrupt lens homeostasis and ultimately result in cataract formation[1–3,5]. So far, no drugs are yet available for the treatment of cataract, and the only method for restoring vision is cataract surgery[1,6]. Despite rapid visual recovery and minimal complications in most cases, globally fewer than one-fifth of patients with

cataract-induced vision impairment or blindness have received surgical intervention due to contraindications, cost, accessibility limitations, and insufficient expertise in developing regions[1–4]. Therefore, the development of affordable pharmacological treatments to reverse cataract is urgently needed.

Lens crystallins are transparent and water-soluble refractive proteins that maintain lens transparency and a high refractive index[1,7]. Crystallin aggregation resulting from mutations, damage, or aging is a key pathogenic mechanism driving cataract development[1,7–10]. Consequently, suppressing crystallin aggregation represents a potential strategy for developing anti-cataract drugs[10]. Lanosterol synthase (LSS, also known as 2,3-oxidosqualene-lanosterol cyclase) is a cyclase that catalyzes the cyclization of (S)−2,3-oxidosqualene to lanosterol and

[1]Department of Infectious Disease, Shenzhen People's Hospital, The Second Clinical Medical College, Jinan University, Shenzhen, China. [2]School of Medicine, Southern University of Science and Technology, Shenzhen, China. [3]Department of Ophthalmology, Shenzhen People's Hospital, The Second Clinical Medical College, Jinan University, Shenzhen, China. ✉e-mail: libin@mail.sustech.edu.cn

plays a crucial role in regulating crystallin aggregation and enhancing lens transparency[8,11–13]. Mutations in the *LSS* gene can induce cataract formation in both animals and humans[8,14]. Additionally, the use of LSS inhibitors has been shown to induce lens clouding in animal models[14–16]. In recent years, it has been reported that the naturally occurring triterpenoid lanosterol and its synthetic analogs can alleviate cataract formation via reducing abnormal aggregation of lens crystallins[8,17–20]. However, the limited solubility and short half-lives adversely impact their therapeutic efficacy against cataracts[21,22].

mRNA represents a new category of nucleic acid drugs and holds enormous potential in protein replacement therapy, regenerative medicine, cancer immunotherapy, and gene editing beyond vaccines[23–29]. mRNA-based therapeutics allow direct cytoplasmic translation of exogenous mRNA, avoiding the risk of insertional mutagenesis caused by transgene integration. Here, we propose that in vitro-transcribed mRNA encoding human LSS (hLSS mRNA) may be leveraged as an alternative anti-cataract agent to augment hLSS and lanosterol levels in the lens and ultimately ameliorate cataract formation. Given the inherent instability and cell membrane impermeability of mRNA molecules[30], we developed a lipid-like material featuring an aromatic scaffold, two secondary amines, and two monounsaturated C18 tails as an ionizable component of lipid nanoparticles (LNPs) for hLSS mRNA delivery. hLSS mRNA was encapsulated into aromatized LNPs through electrostatic interactions between ionizable lipids and mRNA payloads (Fig. 1). After confirming cellular uptake and expression of LNPs-formulated mRNA both in vitro and in vivo, we further investigated its therapeutic efficacy in two rat models of selenium-induced and galactose-induced cataracts (Fig. 1). Our results demonstrated that hLSS mRNA delivered by aromatized LNPs elevated hLSS protein and lanosterol levels in rat lenses following intracameral injection without causing obvious signs of toxicity, ultimately increasing lens transparency in both cataract models.

## Results

### Optimization and characterization of pB-UC18 LNPs

We previously demonstrated that tB-UC18, an ionizable lipid possessing three monounsaturated C18 alkyl chains, enables efficient mRNA delivery to the spleen[31]. In a recent pilot study, we found that its para-substituted counterpart (pB-UC18, Fig. 2a) shows superior ocular mRNA delivery activity when combined with helper lipids DOPE, but it formed microscale particles in the presence of mRNA payloads. Further incorporation of 1.2 mol% DMG-PEG2k not only reduced the hydrodynamic size of the mRNA-lipid complexes from the micrometer to the nanometer scale, but also enhanced their stability for at least 24 days (Supplementary Fig. 1). The optimized lipid nanoparticles (termed pB-UC18 LNPs) had a hydrodynamic diameter ($D_H$) of $173 \pm 6$ nm, polydispersity index (PDI) of $0.17 \pm 0.02$, and zeta potential ($\zeta$) of $11.5 \pm 0.7$ mV (Fig. 2b). The particle size of pB-UC18 LNPs was further validated with transmission electron microscopy (TEM, Fig. 2b).

We subsequently examined the hemolytic activity of pB-UC18 LNPs in vitro. As illustrated in Fig. 2c, pB-UC18 LNPs induced no hemolytic activity at a final mRNA concentration of $2\,\mu g\,mL^{-1}$ under physiological pH, whereas obvious hemolytic activity occurred at acidic pH. These findings indicated that pB-UC18 LNPs were biocompatible at physiological pH. Furthermore, pH-sensitive membrane disruption of red blood cells mediated by pB-UC18 LNPs within the endosomal pH range suggested the endosomal escape potential of pB-UC18 LNPs.

To probe the uptake and endosomal escape of mRNA formulations, cells were treated with pB-UC18 LNPs encapsulating fluorescently labeled mRNA payloads (Cy5-labeled mRNA). After 4 h of treatment, we observed that partial Cy5 fluorescence signals colocalized with the endo-lysosome-specific tracers, while the residual signals were distributed throughout the cytoplasm. These observations

suggested that pB-UC18 LNPs could facilitate the uptake of mRNA payloads and induce effective endosomal escape (Fig. 2d). To further elucidate the mechanism of cellular uptake, we examined the effects of various inhibitors of endocytosis on the uptake efficiency of pB-UC18 LNPs using flow cytometry. As shown in Fig. 2e, three inhibitors, chlorpromazine (CPZ, an inhibitor of clathrin-mediated endocytosis), methyl-β-cyclodextrin (MβCD, an inhibitor of caveolae-mediated endocytosis), and 5-(N-ethyl-N-isopropyl)-amiloride (EIPA, an inhibitor of macropinocytosis), showed significant inhibitory effects on particle uptake, indicating that pB-UC18 LNPs are internalized into cells via all three pathways.

We next designed hLSS mRNA according to its protein sequence (Supplementary Table 1). The coding sequence of hLSS mRNA was shown in Supplementary Table 2. To enhance the stability and translational capacity of mRNA, 5-methoxyuridine (5moU) was introduced into hLSS mRNA during in vitro transcription[32]. The single and clear band shown in the agarose gel validated the purity of 5moU-modified hLSS mRNA (Supplementary Fig. 2). Subsequently, we prepared hLSS mRNA-loaded pB-UC18 LNPs (hereinafter referred to as mLNPs) and tested whether exogenous hLSS mRNA delivered by pB-UC18 LNPs could be successfully translated into corresponding hLSS proteins. Immunofluorescence imaging confirmed elevated LSS levels in mLNPs-treated 293T, HeLa, and SRA01/04 cells (Fig. 2f). Moreover, western blot analysis further indicated that mLNPs induced upregulation of LSS expression (Fig. 2g and h).

### Expression kinetics of pB-UC18 LNPs-formulated mRNA

To explore the effects of ocular delivery routes on mRNA delivery efficiency in vivo, Wistar rats were injected with pB-UC18 LNPs encapsulating firefly luciferase reporter mRNA (FLuc mRNA) at a mRNA dose of 300 ng per eye through intravitreal (IVT), subconjunctival (SCJ), intracameral (IC), or subretinal (SR) injection. FLuc expression was monitored by a small animal imaging system at various time points following ocular delivery. All routes showed ocular FLuc expression at 4 h, but signals declined in IVT, SCJ, and SR groups (Fig. 3a–d). In contrast, the bioluminescence signals in the eyes after IC injection peaked at 26 h and sustained an elevated level at 48 h (Fig. 3a and c), with signals predominantly arising from the lens as evidenced by ex vivo bioluminescence imaging (Fig. 3b and d). Therefore, IC injection was selected for the following in vivo studies. Notably, in comparison with SCJ and IC injection, IVT and SR injection led to higher hepatic signals (Supplementary Fig. 3), indicating unwanted off-target expression for these ocular routes.

Notably, despite being an avascular tissue, the mature lens possesses a unique microcirculatory system that facilitates the transport of mRNA formulations from the anterior chamber to the lens[33,34]. In vivo bioluminescence imaging further demonstrated dose-dependent enhancement of mRNA delivery efficiency and protein expression after IC injection (Fig. 3e and f). At the same dose tested, pB-UC18 LNPs displayed more potent ocular delivery efficiency than clinically approved SM-102 LNPs (Fig. 3e and f).

### Ocular hLSS mRNA delivery mediated by pB-UC18 LNPs

We next sought to replace the FLuc mRNA payloads with functional hLSS mRNA and validate its translation in rat eyes following IC injection of mLNPs. Immunofluorescence analysis indicated that the treated lenses exhibited higher hLSS expression compared to the untreated control (Fig. 4a). The highly expressed hLSS proteins predominantly localized in the eyeball wall (such as corneal stroma, corneal endothelium, and choroid), lens epithelium, and lens cortex rather than in the lens nuclear region (Fig. 4a). The differential expression in the cortical and nuclear regions could be attributed to the special lens structure as lens fiber cells progressively lose organelles and nuclei during differentiation to matain transparency[18,35,36]. Furthermore, liquid chromatography-mass spectrometry (LC-MS) quantitative

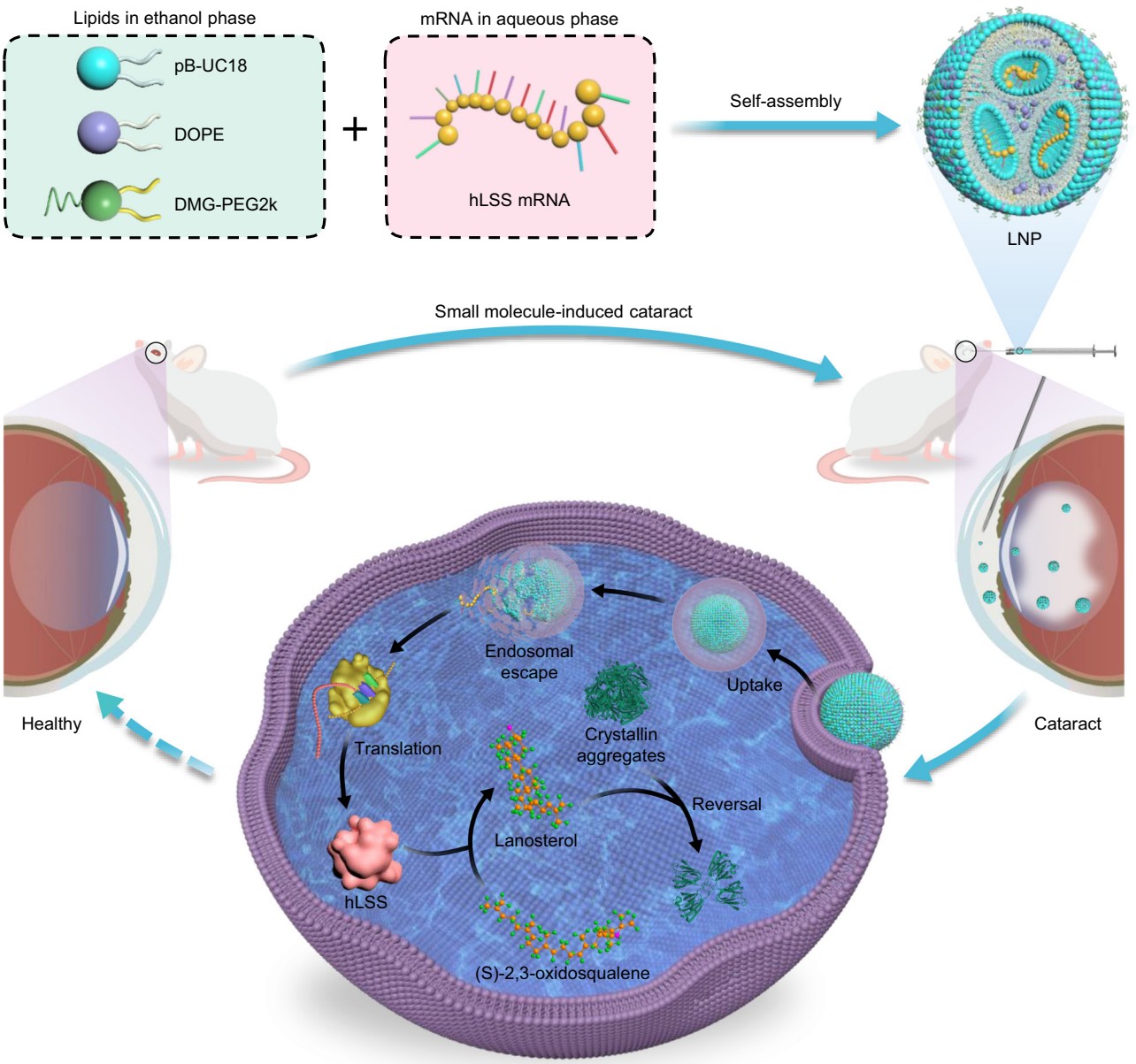

**Fig. 1 | Schematic illustration of hLSS mRNA-loaded pB-UC18 LNPs as an anti-cataract agent.** Three types of lipids, including pB-UC18 (ionizable lipids), DOPE (helper lipids), and DMG-PEG2k (PEGylated lipids), dissolved in ethanol self-assemble into aromatized LNPs in the presence of aqueous mRNA payloads via electrostatic interactions. Exogenous hLSS mRNA delivered by aromatized LNPs can be transported across cell membranes through multiple endocytic pathways and translated into hLSS proteins. mRNA-encoded hLSS proteins facilitate cyclization of (S)−2,3-oxidosqualene to lanosterol, thereby diminishing crystallin aggregation in rat cataract models. pB-UC18, (9Z,9′Z)-N,N′-(1,4-phenylene-bis(methylene))bis(octadec-9-en-1-amine). DOPE, 1,2-dioleoyl-sn-glycero-3-phosphoethanolamine. DMG-PEG2k, 1,2-dimyristoyl-rac-glycero-3-methoxypolyethylene glycol-2000.

analysis indicated that IC injection of mLNPs significantly increased the lanosterol levels in lenses of Wistar rats of different ages (Fig. 4b).

Since the eye is one of the most elaborate organs, we conducted comprehensive safety assessments of mLNPs after IC injection. The cornea and lens were first assessed 24 h after IC injection using a slit-lamp microscope. In comparison with the uninjected group, the sham group and mLNPs group exhibited no significant ocular lesions (including eyelid irritation and redness, corneal opacification, and edema) except for corneal puncture marks (Fig. 4c). Further analysis of histological samples stained with hematoxylin and eosin (H&E) revealed that IC injection of mRNA formulations caused no apparent histological alterations in the eyes (Fig. 4d) and other major organs including the heart, liver, spleen, lungs, and kidneys (Supplementary

Fig. 4), or structural changes in the ocular tissue (Fig. 4d). Besides, there were no statistically significant differences between the uninjected group and mLNPs group in routine blood parameters as well as hepatic and renal function (Supplementary Fig. 5).

## Therapeutic effects of pB-UC18 LSS mRNA-LNPs in cataract models

To evaluate the therapeutic efficacy of mRNA formulations for the treatment of cataracts, we established two rat cataract models: a nuclear cataract model induced by subcutaneous (SC) injection of sodium selenite and a galactose cataract model induced by intraperitoneal (IP) injection of D-galactose. The induction and treatment protocols for the nuclear cataract model were illustrated in Fig. 5a. For

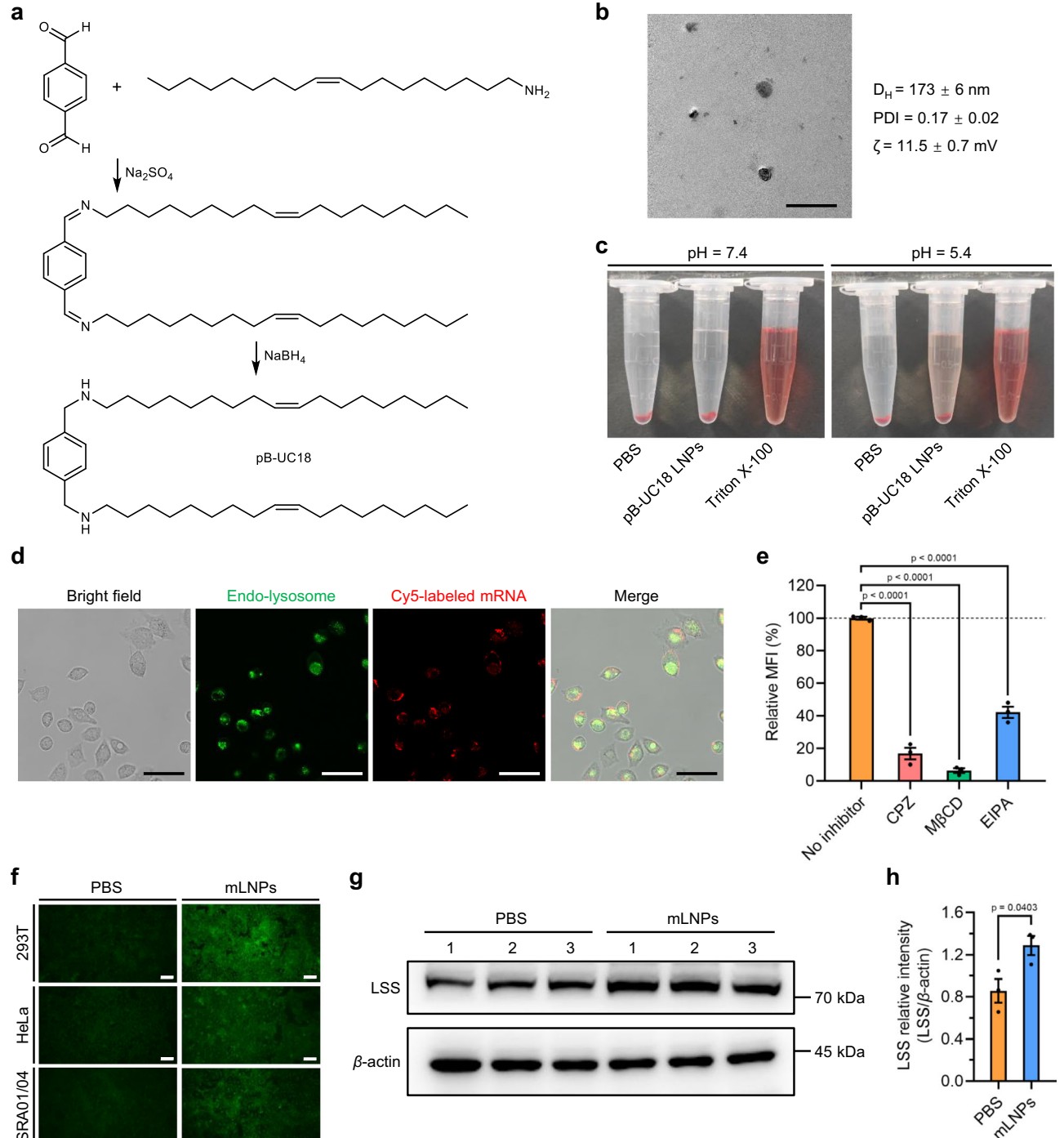

**Fig. 2 | Preparation and characterization of pB-UC18 LNPs. a** Synthetic route of pB-UC18. **b** Representative TEM image (Scale bar, 500 nm) as well as $D_H$, PDI, and ζ of pB-UC18 LNPs determined by dynamic light scattering. **c** Hemolytic activity of pB-UC18 LNPs at pH 7.4 and 5.4. **d** Colocalization analysis of the endosomal escape capacity of pB-UC18 LNPs encapsulating Cy5-labeled mRNA (red). The endolysosomes were specifically stained with green dyes. Scale bar, 50 μm. **e** Effects of different chemical inhibitors of endocytosis on the internalization of pB-UC18 LNPs. The mean fluorescence intensity (MFI) of the no-inhibitor control was defined as 100%. Data are presented as mean ± SEM ($n$ = 3 biological replicates, one-way ANOVA with Dunnett's multiple comparison test). **f** Immunofluorescence images of cells treated with hLSS mRNA-loaded pB-UC18 LNPs for 24 h. Green, LSS proteins. Scale bar, 50 μm. **g** Western blot analysis of the expression level of LSS proteins from cells treated with hLSS mRNA-loaded pB-UC18 LNPs or PBS for 24 h. Endogenous β-actin was used as a loading control. **h** Quantification of the relative intensity of LSS protein expression from **g**. Data are presented as mean ± SEM ($n$ = 3 biological replicates, unpaired, two-tailed Student's $t$ test). Source data are provided as a Source Data file.

all treated groups, one eye of each rat was treated with empty LNPs (eLNPs) or mLNPs through the anterior chamber, while the contralateral eye was left untreated for comparisons. Slit-lamp examination and ex vivo visual inspection of lens revealed that there was a remarkable improvement in lens turbidity in the injected eye

compared to the uninjected contralateral eye for the mLNPs-treated cataract model group (Fig. 5b and c). A further analysis of the cataract stage confirmed that cataract development in the injected eye was significantly lower than the uninjected contralateral eye (Fig. 5d). Nevertheless, eyes in the healthy control group (without injection of

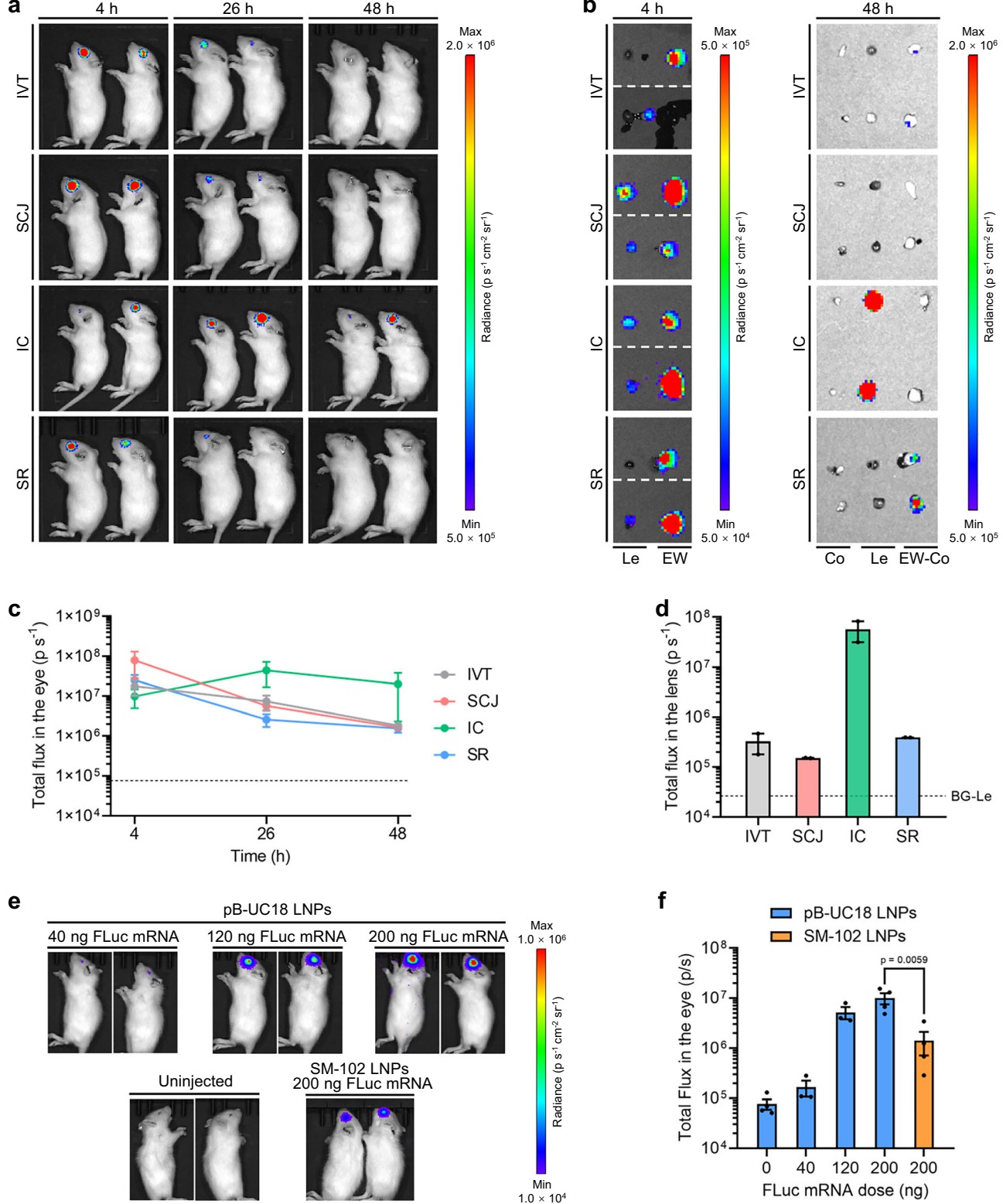

**Fig. 3 | Expression kinetics of FLuc mRNA delivered by pB-UC18 LNPs.**
**a** Representative whole-body bioluminescence images of Wistar rats at different times after IVT, SCJ, IC, or SR injection of FLuc mRNA-loaded pB-UC18 LNPs (300 ng FLuc mRNA per injection). **b** Ex vivo bioluminescence images of the ocular tissues from **a**. Le, lens. EW, eyeball wall. Co, cornea. EW-Co, eyeball wall without cornea. Dashed lines indicated the border between two images. Following 4 h whole-body bioluminescence imaging, 2 out of 4 rats from each group were sacrificed for ex vivo imaging. The remaining 2 rats were subjected to whole-body bioluminescence imaging repeatedly over time (26 h and 48 h) and sacrificed at 48 h for ex vivo imaging. **c**, **d** Quantification of luminescence flux in eyes (**c**) and lenses (**d**), which

were from **a** and **b** (48 h), respectively. Dashed lines in (**c**) indicated the background flux in the eyes. BG-Le in (**d**) indicated the background flux in the lenses.
**e** Representative whole-body bioluminescence images of Wistar rats 4 h after IC injection of FLuc mRNA-loaded pB-UC18 LNPs at different mRNA doses. Uninjected rats and rats injected with SM-102 LNPs at a FLuc mRNA dose of 200 ng served as control groups. **f** Quantification of luminescence flux in the eyes from **e** ($n = 4$ rats for 0 ng and 200 ng; $n = 3$ rats for 40 ng and 120 ng; one-way ANOVA with Sidak's multiple comparison test). For all relevant panels, data are presented as mean ± SEM. Source data are provided as a Source Data file.

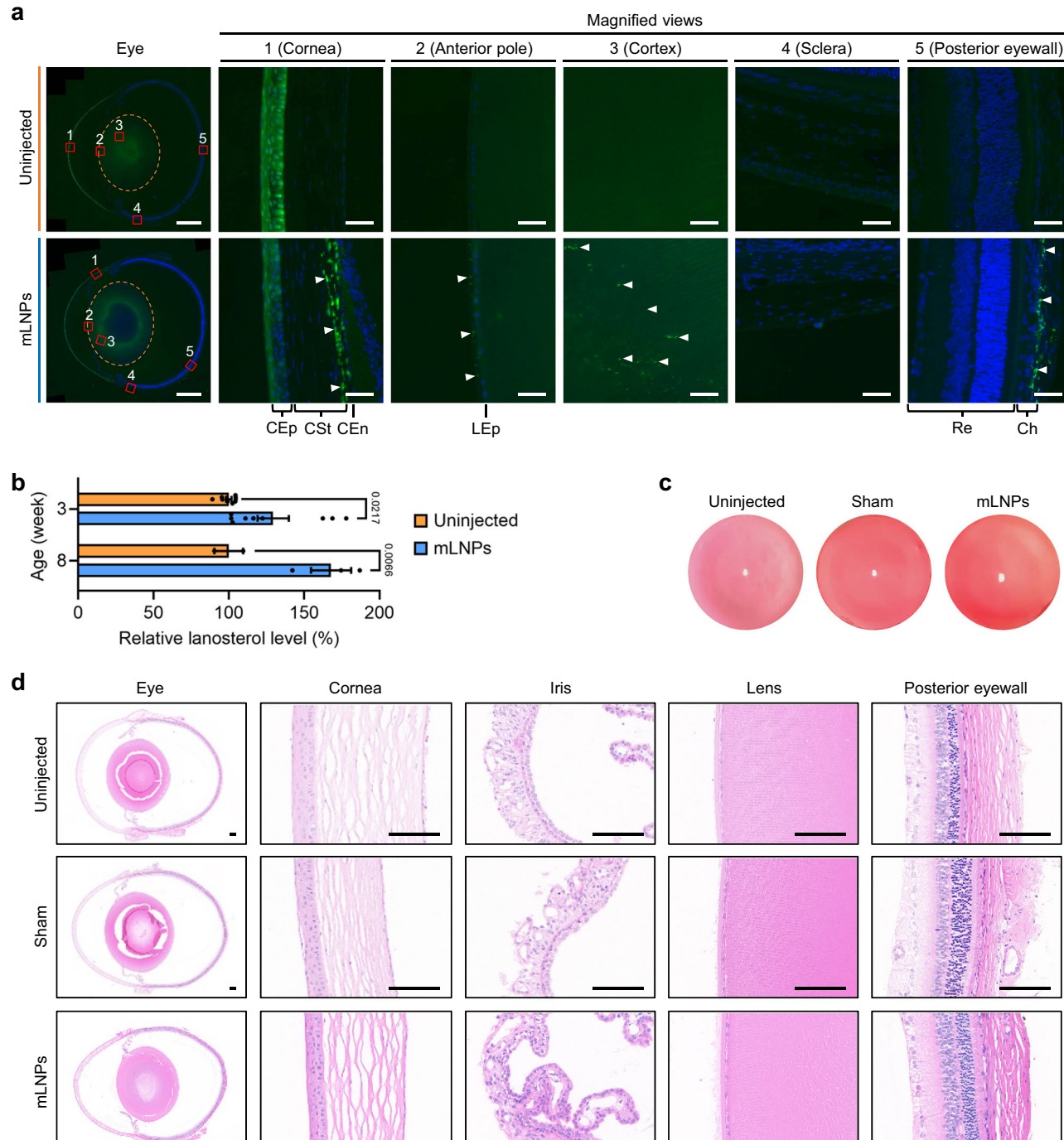

**Fig. 4 | Ocular hLSS expression, lanosterol levels, and in vivo biosafety assessments after treatment with mLNPs. a** Immunofluorescence images of ocular sections of three-week-old rats from the uninjected and mLNPs groups. The region within the orange dashed ellipse indicated the lens. Scale bar, 1 mm. Magnified views of the detailed ocular zones (red boxes) were shown in the right panel. CEp, corneal epithelium, CSt, corneal stroma, CEn, corneal endothelium, LEp, lens epithelium, Re, retina, Ch, choroid. Representative fluorescence signals were indicated by white triangles. Scale bar, 40 μm. **b** LC-MS measurements of the lanosterol levels in the lenses of mLNPs-treated rats aged 3 weeks (3 w) and 8 weeks (8 w). The lanosterol levels in the lenses of uninjected age-matched rats were defined as 100%. Data are presented as mean ± SEM (*n* = 3 biological replicates for

Uninjected_3 w and mLNPs_3 w, each biological replicate consisting of 3 technical replicates; *n* = 2 biological replicates for Uninjected_8 w; *n* = 3 biological replicates for mLNPs_8 w; two-way ANOVA with two-sided Sidak's multiple comparison test with adjustment). **c** Ocular surface slit-lamp photographs of seven-week-old rats from the uninjected, sham, and mLNPs groups. **d** Images of H&E-stained ocular sections (including eye, cornea, iris, lens, and posterior eyewall) of seven-week-old rats from the uninjected, sham, and mLNPs groups. Scale bar, 100 μm. For all relevant panels, eyes of Wistar rats from the mLNPs group were subjected to analysis 24 h after IC injection of mLNPs at a hLSS mRNA dose of 300 ng per injection. Uninjected or sham eyes served as a control group. Source data are provided as a Source Data file.

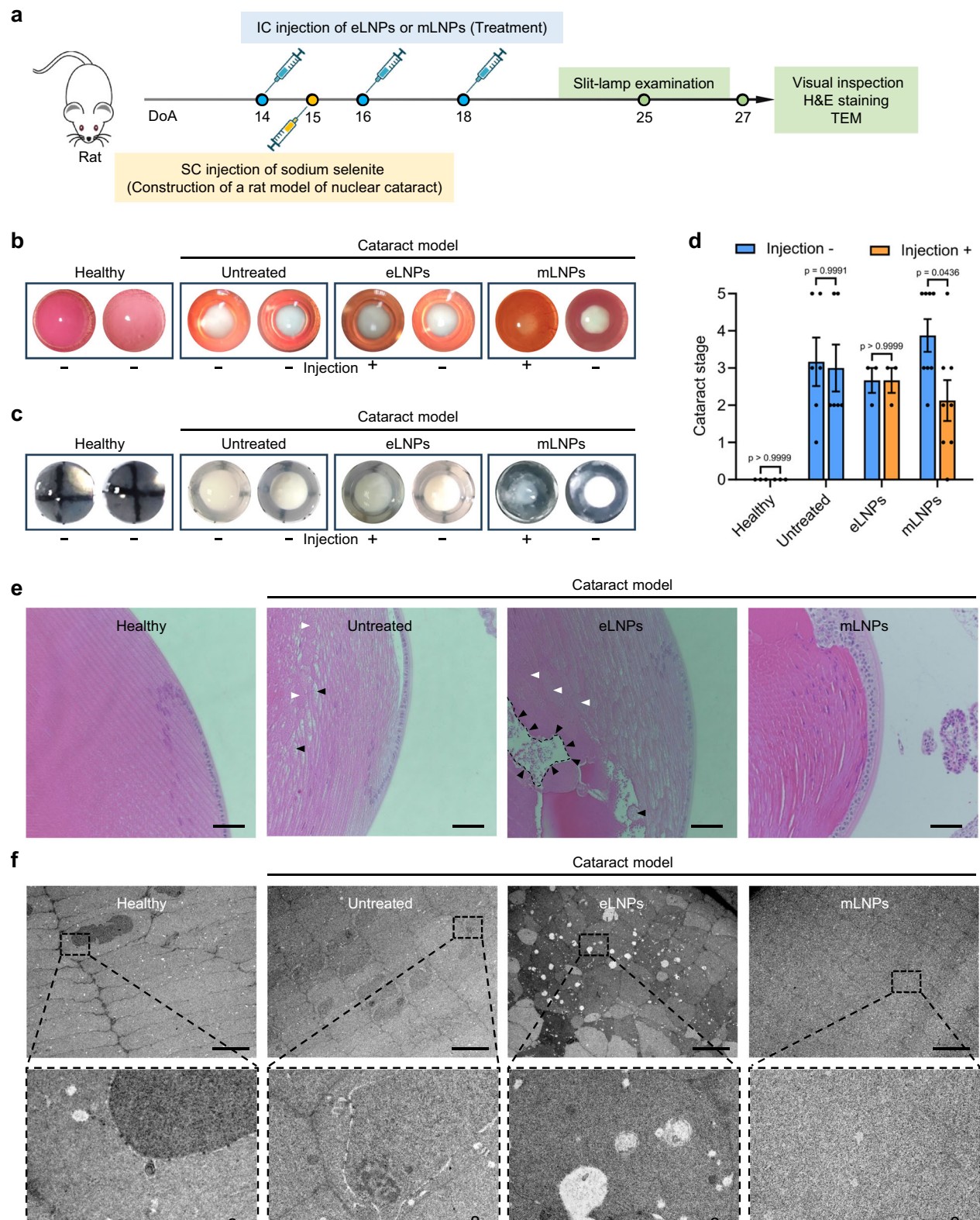

**Fig. 5 | Therapeutic effects of mLNPs in a sodium selenite-induced nuclear cataract model. a** Experimental workflow for the induction and treatment of nuclear cataract in rats. DoA, days of age. **b, c** Slit-lamp photographs of eyes (**b**) and visual inspection of ex vivo lenses (**c**) from the healthy group and three groups of nuclear cataract rats (untreated, eLNPs, and mLNPs groups) taken at 25 and 27 days of age, respectively. For the eLNPs and mLNPs groups, one eye of each rat was treated with eLNPs or mLNPs (injection +) while the contralateral eye from the same rat was left untreated (injection -) for comparisons to eliminate the rat-to-rat intergroup variability. **d** Cataract stages at the experimental endpoint. Data are presented as mean ± SEM ($n = 3$ rats for the healthy group; $n = 6$ rats for the untreated group; $n = 3$ rats for the eLNPs group; $n = 8$ rats for the mLNPs group; one-way ANOVA with Sidak's multiple comparison test). **e** Representative H&E-stained lens sections. Representative vacuoles and irregular cellular arrangements in the untreated and eLNPs groups were indicated by black and white triangles, respectively. Scale bar, 50 μm. **f** Representative TEM images of lenses. Magnified views of the detailed zones (dashed boxes) in the top panel were shown in the bottom panel. Scale bar, 10 μm.

sodium selenite), untreated cataract model group (without injection of formulations), and eLNPs-treated cataract model group exhibited comparable cataract grades (Fig. 5d).

H&E staining and TEM ultrastructural analysis reflected regular arranged lens fiber cells with no obvious vacuoles in the healthy group (Fig. 5e and f). However, irregular cellular arrangements and considerable vacuoles within the lens cortex were observed for the untreated and eLNPs cataract model groups (Fig. 5e and f). In contrast, these alterations were rare within the lens for the mLNPs cataract model group.

We proceeded to assess the therapeutic efficacy of mRNA formulations in the galactose-induced cataract model (Fig. 6a). As shown in Fig. 6b, repeated intraperitoneal injection of D-galactose successfully induced cataract formation. In other words, annular equatorial-like vesicles or radial turbidity appeared in the lens cortex of groups at 54 days of age (T + 0). Following administration of mLNPs, the relative turbidity area including annular equatorial-like vesicles and radial turbidity in the lenses diminished over time, as evidenced by the slit-lamp photographs and the slopes of linear regression (Fig. 6b and c). Pathological analysis further confirmed the obvious remission of lens fiber swelling (Fig. 6d). Terminal deoxynucleotidyl transferase dUTP nick end labeling (TUNEL) staining revealed dramatically decreased apoptotic cell death in the lens epithelium and cortex (Fig. 6e), which suggested that the protection of lens cells from apoptosis via modulation of LSS-mediated pathway might contribute to reversal of protein aggregation in cataracts.

Taken together, the therapeutic effects observed in both cataract models demonstrated that ocular delivery of in vitro-transcribed hLSS mRNA via lipid nanoparticles was able to ameliorate both nuclear and galactose cataract in rats.

## Discussion

Previous studies have demonstrated that lanosterol, an intermediate in cholesterol biosynthesis, plays a crucial role in the maintenance of lens transparency[8,11]. The use of lanosterol and other sterol derivatives to inhibit cataract formation has been extensively investigated in pre-clinical studies[8,11–13,37]. For instance, injection of lipid-polymer hybrid nanoparticles loaded with lanosterol into the vitreous cavity could attenuate cataract in dogs[8]. Lanosterol-loaded thermogel administered by SCJ injection has been reported to reduce crystallin aggregation in cynomolgus monkeys with cortical cataract[37]. However, the limited solubility and short half-life of lanosterol hinder its therapeutic efficacy[21,22]. In addition, some studies demonstrated that in vitro incubation of lenses with lanosterol solution or lanosterol liposomes did not reverse lens opacification[38,39]. More recently, a genetic association study has revealed that there was no significant correlation between lanosterol and cataract risk, potentially due to the differential effects of lanosterol on cataract subtypes[40].

As a new class of drugs, mRNA has gained more and more attention for the prevention and treatment of multiple refractory diseases[23–29]. In recent years, advanced ocular mRNA delivery systems have expanded its applications to ocular diseases such as corneal inflammation and proliferative vitreoretinopathy[41–43]. In this study, we reported the use of hLSS mRNA-loaded pB-UC18 LNPs as anti-cataract mRNA formulations for cataract treatment. hLSS mRNA delivered to the lens by aromatized LNPs via IC administration enabled translation of exogenous mRNA into hLSS proteins and indirect elevation of the endogenous lanosterol in the lenses in soluble and persistent forms. Injection of the resulting hLSS mRNA-LNPs into the anterior chamber of eyes could effectively attenuate cataract severity without noticeable adverse effects in two rat cataract models mimicking age-related and diabetic cataracts. Together, from the lanosterol synthase perspective, this work demonstrates that ocular delivery of mRNA-encoded LSS enables reversal of lens opacification, possibly through the elevation of hLSS proteins and lanosterol contents, along with modulation of LSS-

mediated antiapoptotic pathways in the lenses, offering potential avenues for the treatment of cataract.

Despite showing the potential to alleviate cataract formation, this study has limitations that merit consideration. First, ocular delivery provides an alternative option to avoid complex surgical procedures and devices, but still requires ophthalmic surgical skills. Second, minimally invasive or non-invasive ocular delivery systems remain to be explored because improper ocular injection technique might cause ocular complications or even aggravate cataract. Another limitation is that the anti-cataract efficacy of hLSS mRNA-LNPs is only evaluated in rat models. Further work is needed to explore their utility and cross-species translatability in non-human primates.

## Methods

### LNP formulations

The purity of 5moU-modified hLSS mRNA (APExBio) was determined by 1% native agarose gel before encapsulation. LNP formulations were prepared through the previously reported method[31]. Briefly, pB-UC18 was synthesized via the condensation reaction followed by reduction of the Schiff base intermediate[31]. pB-UC18, along with helper lipids DOPE (Avanti) and PEGylated lipids DMG-PEG2k (TargetMol), were dissolved in ethanol at a molar ratio of 40:40:0 or 40:40:1 to form the ethanol phase. Meanwhile, mRNA was diluted with phosphate-buffered saline (PBS) to obtain the aqueous phase. The ethanol phase and mRNA aqueous phase were then rapidly mixed at an N:P ratio of 2:1 and a volume ratio of 1:9 and allowed to stand for 10 min prior to use.

### Particle size and zeta potential

The particle size and zeta potential of LNPs stored at ambient temperature for different time periods were measured using dynamic light scattering (Brookhaven, 90Plus PALS) and analyzed using the Particle Solution software. The particle size of freshly prepared pB-UC18 LNPs was also assessed by transmission electron microscopy (Thermo Fisher Scientific, Talos L120C).

### Hemolysis assay

Blood samples from Wistar rats were collected with EDTA-coated anticoagulant tubes. Red blood cells (RBCs) were isolated via centrifugation at $95\,g$ for 10 min, washed, and suspended in PBS with pH 5.4 and 7.4. hLSS mRNA-loaded pB-UC18 LNPs at a mRNA concentration of $4\,\mu g\,mL^{-1}$ were then incubated with an equal volume of RBCs (4%, v/v) for 1 h at 37 °C and centrifuged ($95\,g$) for 10 min at ambient temperature for imaging. RBCs lysed with 1% Triton X-100 were used as a positive control, while RBCs treated with PBS served as a negative control.

### Cellular uptake and endosomal escape

SRA01/04 cells obtained from Dr. Jun Zhao's lab were seeded at a density of $1.5 \times 10^4$ cells per well in a 96-well plate. Following overnight culture, cells were pre-incubated with either PBS or inhibitors including CPZ (MedChemExpress, $10\,\mu g\,mL^{-1}$ final concentration), MβCD (MedChemExpress, 5 mM final concentration), and EIPA (MedChemExpress, $10\,\mu g\,mL^{-1}$ final concentration) for 30 min at 37 °C and then exposed to pB-UC18 LNPs containing Cy5-labeled mRNA (APExBio) at a final mRNA concentration of $2\,\mu g\,mL^{-1}$ for 4 h. After washing twice with PBS, cells were subjected to flow cytometric analysis (Agilent, Novo-Cyte 2000R) with the NovoExpress software. Gating strategy depicted in Supplementary Fig. 6. For confocal microscopy imaging, HeLa cells (C6330, Beyotime) were seeded at a density of $4 \times 10^4$ cells per confocal dish (35 mm, Biosharp). Following overnight culture, cells were exposed to pB-UC18 LNPs containing Cy5-labeled mRNA at a final mRNA concentration of $2\,\mu g\,mL^{-1}$ for 4 h, stained with Lyso-Tracker Green (Beyotime), and imaged with confocal microscopy (Leica, TCS SP8).

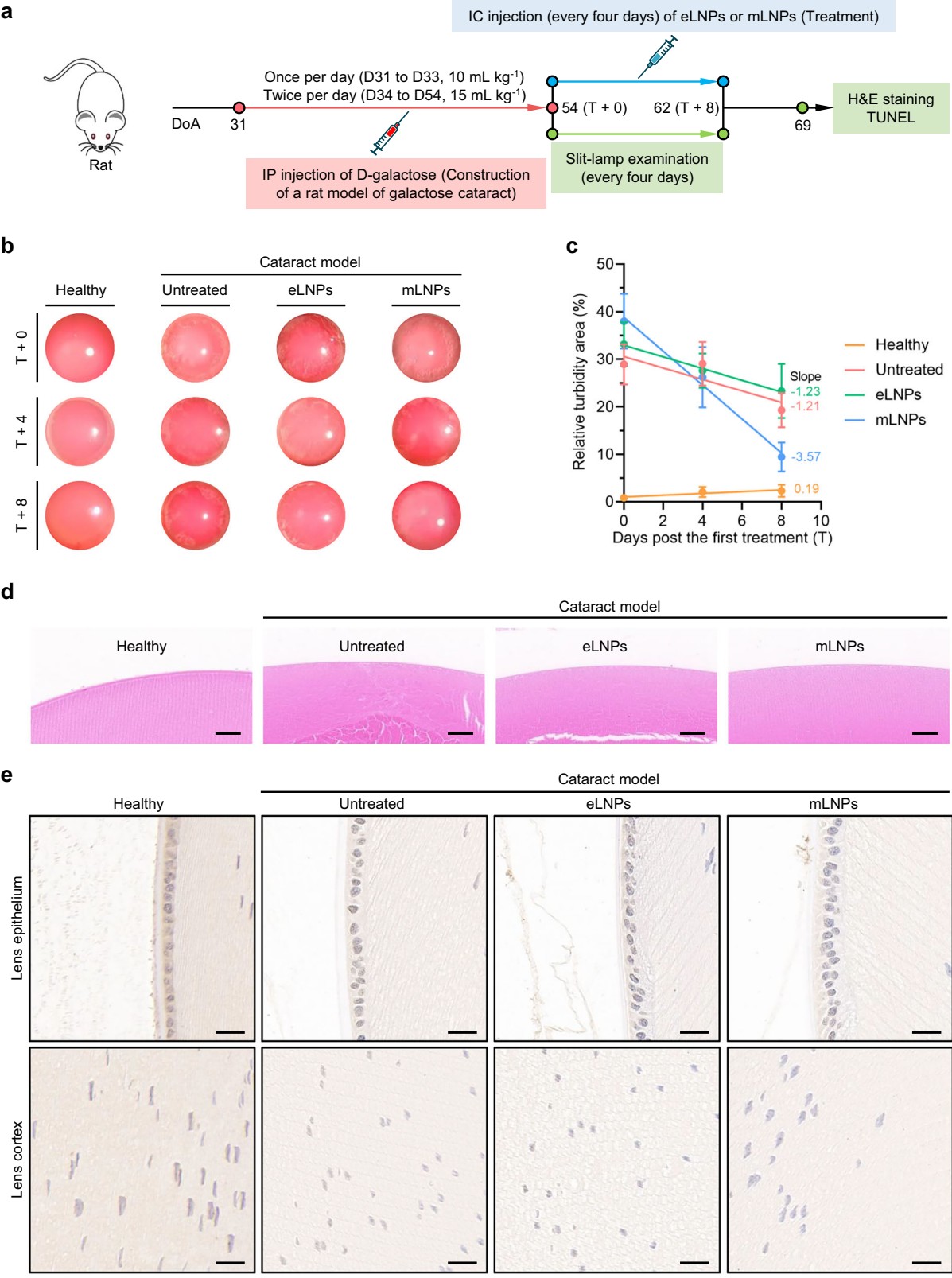

**Fig. 6 | Therapeutic effects of mLNPs in a galactose-induced cataract model.**
**a** Experimental workflow for the induction and treatment of galactose cataract in rats. DoA, days of age. **b** Slit-lamp photographs of eyes from the healthy group and three groups of galactose cataract rats (untreated, eLNPs, and mLNPs groups) at days 0 (T + 0), 4 (T + 4), and 8 (T + 8) post the first treatment (T). **c** Quantification of the relative turbidity area in the lenses for the healthy, untreated, eLNPs, and mLNPs groups. The turbidity area was normalized to that of the corresponding lens. Data are presented as mean ± SEM ($n = 8$ lenses for the healthy group; $n = 8$ lenses for the untreated group; $n = 8$ lenses for the eLNPs group; $n = 9$ lenses for the mLNPs group; one-way ANOVA with Dunnett's multiple comparison test. It was noted that one rat died in the healthy group on day T + 8 owing to slit-lamp examination under anesthesia). **d** Representative H&E-stained lens sections. Scale bar, 50 µm. **e** Representative TUNEL-stained histological images of lens epithelium and lens cortex. Blue, normal nuclei. Brown, apoptotic nuclei. Scale bar, 20 µm.

## Immunofluorescence analysis of cellular LSS Expression

293T (C6008, Beyotime), HeLa, and SRA01/04 cells were seeded in 96-well plates at densities of $2 \times 10^4$, $1 \times 10^4$, and $1.5 \times 10^4$ cells per well, respectively, and allowed to culture overnight. Cells were then treated with pB-UC18 LNPs containing 200 ng of hLSS mRNA. Cells treated with PBS served as a control. Twenty-four hours post-treatment, cells were fixed by ethanol at −20 °C following supernatant removal, permeabilized with Triton X-100 (0.5%, v/v), and blocked with a bovine serum albumin solution (Servicebio, 5%, w/v) for 30 min. Cells were then incubated with LSS polyclonal antibody (Proteintech, Cat No. 18693-1-AP, 1:100) at room temperature for 90 min and FITC goat anti-rabbit IgG (Abclonal, Cat No. AS011, 1:100) at room temperature for 90 min and imaged using a fluorescence microscope (ZEISS, Vert.A1).

## Western blot analysis of cellular LSS level

293T cells were seeded at a density of $1.2 \times 10^5$ cells per well in a 24-well plate and allowed to culture overnight. Cells were then treated with pB-UC18 LNPs containing 1200 ng of hLSS mRNA for 24 h at 37 °C. Subsequently, cells were harvested and suspended in RIPA buffer supplemented with PMSF (Beyotime, 1 mM final concentration) for 60 min at 4 °C. The protein concentration in the lysate was quantified using a BCA protein assay kit (Beyotime). Samples were then diluted with loading buffer, heated for 10 min at 100 °C, and separated by sodium dodecyl sulfate-polyacrylamide gel electrophoresis. After electrophoresis, proteins were then transferred onto a polyvinylidene difluoride membrane blocked with blocking buffer. Subsequently, the membranes were probed with antibodies against LSS (Proteintech, Cat No. 18693-1-AP, 1:500) or $\beta$-actin (Beyotime, Cat No. AA128, 1:7000) in blocking buffer for 1 h at 37 °C and washed three times with washing buffer. The blots were then incubated with horseradish peroxidase-conjugated goat anti-rabbit IgG (Abclonal, Cat No. AS014, 1:7000) or horseradish peroxidase-conjugated goat anti-mouse IgG (Beyotime, Cat No. A0216, 1:7000) for 1 h at room temperature, washed three times with washing buffer, incubated with the western blotting substrate (Beyotime), and imaged with an imaging system (Thermo Fisher Scientific, iBright FL1000). Quantification of Western blot bands was done by densitometry using the ImageJ software.

## In vivo bioluminescence imaging

All animal procedures were approved by the Institutional Animal Care and Use Committee of the First Affiliated Hospital of Southern University of Science and Technology (protocol # 20200330-16) and conducted in accordance with the guidelines for the care and use of laboratory animals. Wistar rats were housed under standard conditions (12 h light and 12 h dark cycles, 20–22 °C, 40–60% humidity) with food and water provided ad libitum. Three-week-old Wistar rats were anesthetized using isoflurane before mydriasis with eyedrops containing 0.5% phenylephrine and 0.5% tropicamide and injected with 1.5 μL of FLuc mRNA-loaded pB-UC18 LNPs at a FLuc mRNA (APExBio) dose of 300 ng per injection through IVT, SCJ, IC, or SR injection. At 4 h, 26 h, or 48 h post-injection, D-luciferin potassium salt was administered intraperitoneally at a dose of 125 mg kg$^{-1}$. Ten minutes later, rats were anesthetized with isoflurane, imaged using a small animal imaging system (PerkinElmer, Lumina LT) and analyzed using the Living Image software. At 4 h and 48 h post-injection, the eyeballs were further harvested and dissected into the lenses and other tissues for ex vivo imaging. In a separate experiment, the eye, heart, liver, spleen, lungs, and kidneys were harvested 4 h post-injection for ex vivo imaging. For the dose response study, Wistar rats receiving 1.5 μL of FLuc mRNA-loaded pB-UC18 LNPs at different mRNA doses (40, 120, and 200 ng per injection) through IC injection were imaged at 4 h post-injection. FLuc mRNA-loaded SM-102 LNPs prepared at a SM-102:FLuc mRNA weight ratio of 10.75:1 and a SM-102 (Macklin):DSPC (Avanti):cholesterol (Sangon Biotech):DMG-PEG2k (TargetMol) molar ratio of 50:10:38.5:1.5 were used as a positive control[44].

## Immunofluorescence analysis of LSS expression in ocular tissues

Three-week-old Wistar rats were anesthetized using isoflurane before mydriasis with eyedrops containing 0.5% phenylephrine and 0.5% tropicamide and injected with 1.5 μL of mLNPs at a hLSS mRNA dose of 300 ng per injection through IC injection. Eyes from untreated rats served as a control. At 24 h post-injection, ocular tissues were harvested for preparation of paraffin-embedded eye tissue sections using the standard procedures. Slices were then blocked with 5% BSA at room temperature for 30 min after antigen retrieval with TE buffer (pH 9) and incubated overnight at 4 °C with LSS polyclonal antibody (Proteintech, Cat No. 18693-1-AP, 1:100) and Alexa Fluor 488-labeled goat anti-rabbit IgG (Servicebio, Cat No. GB25303, 1:400) at room temperature for 50 min. Slices were further stained with a DAPI staining solution (Servicebio) for 10 min at room temperature, rinsed three times with PBS, and incubated with Tissue Autofluorescence Bursting Agent (Servicebio) in droplet form for 5 min. After sealing, slices were imaged using a digital slide scanner (3DHISTECH, Pannoramic MIDI).

## Quantitative analysis of lanosterol levels in lenses via LC-MS

Wistar rats aged 3 or 8 weeks were anesthetized and intracamerally injected with 1.5 μL of mLNPs according to the above protocol. Rats were then euthanized 24 h post-injection, and the lenses were excised and weighed. Subsequently, 20 μL of lanosterol analog, cholesterol-d7 (1.0 mg mL$^{-1}$, served as an internal standard), was added to lenses, which was then homogenized in 600 μL of chloroform/methanol mixture (v:v, 1:2). Then, 300 μL of chloroform and 450 μL of saturated potassium chloride solution were added. The mixture was vortexed for 5 min and centrifuged at 3500 $g$ for 5 min. The upper aqueous phase was then extracted twice with chloroform (300 μL each), combined with the previous lower organic phase, and evaporated under a nitrogen stream. Samples were resuspended in 100 μL of methanol for LC-MS quantitative analysis (SCIEX, 4000 QTRAP). The operating conditions for LC-MS were as follows: the ion source, atmospheric pressure chemical ionization, was set to run in the positive ion mode; the chromatographic column (1.8 μm, Agilent) with dimensions 100 × 4.6 mm was kept at 40 °C; the injection volume was 5 μL; chromatographic separation was done under isocratic conditions, with a mobile phase of isopropanol/acetonitrile/water/formic acid (48.75:48.75:2.4975:0.0025%, v/v/v/v) and flow rate of 0.8 mL min$^{-1}$.

## In vivo biosafety evaluation

Seven-week-old male Wistar rats receiving 1.5 μL of intracamerally injected mLNPs according to the above protocol were subjected to ocular slit-lamp (Bolan, BL-66B) examination 24 h post-injection. Uninjected rats and rats receiving a sham procedure (cornea puncture only) were included as two control groups. Blood samples from different groups were collected for measuring routine blood parameters and hepatic and renal function. Meanwhile, the major organs, including the eye, heart, liver, spleen, lungs, and kidneys, were harvested for H&E staining using the standard procedures.

## Reversal of selenite-induced cataracts in Wistar rats

Wistar rats with eyes open at approximately 14 days of age were randomly divided into four groups: the healthy group (without injection of sodium selenite), the untreated cataract model group (without injection of formulations), the eLNPs-treated cataract model group, and the mLNPs-treated cataract model group. At 14 days of age, one eye of each rat was intracamerally injected with eLNPs or equivalent mLNPs containing 200 ng of hLSS mRNA, while the contralateral eye from the same rat was left untreated for all treated groups. At 15 days

of age, the selenium-induced cataract model was established by subcutaneous injection of sodium selenite solutions at a dose of 3.14 mg kg⁻¹. At 16 and 18 days of age, the same eyes of rats were further injected with the above dose of eLNPs or mLNPs through the anterior chamber. At 25 days of age, rats underwent slit-lamp examination to assess the stage of cataract development according to the cataract classification[45]. At 27 days of age, rats were euthanized, and their lenses were photographed, fixed, and subjected to H&E staining and transmission electron microscopy (Hitachi, HT7800).

**Reversal of galactose-induced cataracts in Wistar rats**

The galactose-induced cataract model was established by intraperitoneal injection of D-galactose solutions (50%, w/v, Aladdin) into Wistar rats once daily at a dose of 10 mL kg⁻¹ for three days (from 31 to 33 days of age) and twice daily at a dose of 15 mL kg⁻¹ for the following three weeks (from 34 to 54 days of age). At 54 days of age, rats were randomly divided into three groups: untreated cataract model group, eLNPs-treated cataract model group, and mLNPs-treated cataract model group. Age-matched rats without injection of both D-galactose and formulations served as the healthy control. For all treated groups, eLNPs or equivalent mLNPs containing 300 ng of hLSS mRNA were injected into the anterior chamber of two eyes every four days (from 54 to 62 days of age). Meanwhile, slit-lamp examination was conducted every four days (from 54 to 62 days of age). The area of each lens and the turbidity area within the lens were calculated using the ImageJ software. At 69 days of age, ocular tissues were harvested for H&E and TUNEL staining using the standard procedures.

**Reporting summary**

Further information on research design is available in the Nature Portfolio Reporting Summary linked to this article.

## Data availability

Source data are available for Figs. 2b, e, g, h, 3c, d, 3f, 4b, 5d, and 6c and Supplementary Figs. 1, 2, and 5 in the associated source data file. Source data are provided with this paper.

## Code availability

The authors declare no code availability.

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

## Acknowledgements

This study was supported by the Shenzhen Science and Technology Program (Grant Nos. RCYX20200714114539061 to B.L. and GJHZ20220913142618036 to J.Z.).

## Author contributions

B.L. conceived the study and supervised the project. R.S. performed most of the experiments and data analysis. Y.L., M.Z., Z.L., R.Z., and J.Z. assisted with animal experiments. B.L. and R.S. wrote and finalized the manuscript. All authors provided critical feedback on the research, analysis, and manuscript.

## Competing interests

B. Li, R. Song, Y. Lin, and Z. Liu are listed as inventors on a patent application describing the use of lipid nanoparticles as a platform for hLSS mRNA delivery and cataract treatment. The other authors declare no competing interests.
