## [Transparent Peer Review file · Nature Communications]

Ocular delivery of lipid nanoparticles-formulated mRNA encoding lanosterol synthase ameliorates cataract in rats

Corresponding Author: Professor Bin Li

Version 0:

Reviewer comments:

Reviewer #1

(Remarks to the Author)

Ocular delivery of lipid nanoparticle-formulated mRNA encoding lanosterol synthase ameliorates cataract in rats
Song et al

The manuscript describes the use of a lipid nanoparticle carrying an mRNA encoding copy of human lanosterol synthase. The first part focuses on the production and optimisation of the delivery mechanism, while the second part focuses on its use in two mice models for cataract development. The work described seems carefully planned and executed, with relevant controls in all steps. As a geneticist, I cannot comment in depth on the development of the nanoparticle, but it appears appropriate and logical. The testing part is well planned and the results appear mostly convincing.

I do not have any major comments for the work but a few extra checks and further explanation at specific points will help.

1. In Figure 3 the description of d seems to interrupt the description of c
2. In Figure 5 d the mLNPs treated lens still seems to carry cataract changes which I am not sure agree with the statement that "the lens fiber cells in the healthy group exhibited a regular arrangement with no discernible gaps between cell nuclei (Figures 5d and 5e)". Could the author be a bit more descriptive by adding an arrow or other indication of what they comment on?
3. The discussion is a bit short and does not comment on more recent findings looking at the use of lanosterol to restore lens clarity. Although, older work is cited such as Zhao et al (2015), More et al (2006) and Funk et al (2005), studies such as Shanmugam et al (2015), Daszynski (2019) and Hashimi et al (2024) are not included. Will it be possible for the authors to comment on how this study compares with these recent developments in their discussion and what that means for the use of lanosterol synthase as a treatment for cataract?
4. In Figure S8, if the error bar represents the standard error of the mean, how is the test non-statistically significant with no error bar overlap? Also, should the authors test the mLNPs treated group (the focus of the work) against the other groups instead of comparing the healthy group to all others?

Reviewer #2

(Remarks to the Author)

Ocular delivery of lipid nanoparticle-formulated mRNA encoding lanosterol synthase ameliorates cataract in rats

It is estimated that over 95 million people suffer from cataracts globally. While cataract replacement surgery exists, many people don't have access to this surgery. Thus, novel therapeutics are still desired. The authors demonstrate that an LNP-mRNA therapeutic expressing human lanosterol synthase can reduce cataracts in two rat models. This article is well-written. There is a good use of experimental controls. Data in Figure 5 with the sodium selenite-induced nuclear cataract model appears convincing.

However, there are major concerns. Please see all the comments listed below.

(Minor) Figure 1. It looks like a human eye is used in the diagram instead of a rodent eye. Please switch out to a rodent eye, with rodent sized lens.

Figure 2E. The authors claim pB-UC18 LNPs are mainly internalized into cells via clathrin-mediated endocytosis and micropinocytosis, but fail to discuss if this makes sense for lens internalization. Would cells in the lens use these mechanisms? Please add in the discussion.

Figure 2F&G. Data in F and G do not align. There is very little LSS protein observed in PBS treated cells (F), but in G there is a strong WB band. Which cells were used for the WB analysis? Also, authors could add the WB quantification to this figure instead of placing it in the supplemental figures.

(Major) Figure 4: Authors state: "Tissue immunofluorescence analysis indicated the treated lenses exhibited higher levels of expression of hLSS proteins compared to the untreated group after formulation administration (Figure 4a)." The authors then go on to say, "The expression of hLSS proteins predominantly localized in the eyeball wall (such as corneal stroma, corneal endothelium, and choroid), the lens epithelium, and lens cortex rather than the nuclear region of the lens 24 h after a single AC injection of mLNPs (Figure 4a)."

I don't see data that supports these statements. I don't see any expression in the lens epithelium, very little in the cortex, and why would there be any expression in the choroid after an AC injection? Also, the authors are expressing human LSS, is that different than rat LSS? Does the antibody detect both rat and human versions? Would it be helpful to distinguish between endogenous levels and the overexpressed levels from the LNPs? The authors should clarify. Overall, there is very little evidence that the LNPs transfect and express mRNA in the lens. The most convincing data is the luciferase data, but luciferase data often results in false positives and must be supported by immunofluorescence or WB data. It did look like there was expression in the cornea. Could that then lead to increased HPLC levels? Immunofluorescence data needs to be more convincing to claim that the LNPs transfect the lens.

Figure 5: These data are convincing. What are your statistics for these data, how many animals, how many replicates? In the discussion, please discuss how effective this treatment is compared to others. How translatable is this technology? How different is this model in comparison to the ways most patients get cataracts (genetics, age, trauma, etc)?

Figure 6: This model is less convincing and so are improved outcomes.

(Minor) Figure S3: Which figure is this dataset referring to? Please clarify.

Figure S7: Is there any significant difference between the orange bars between groups? It looks like you are getting a significant difference in cataract stage in mLNPs because injection – group has a worse cataract stage (increased compared to untreated and eLNPs). What are the sample sizes?

Version 1:

Reviewer comments:

Reviewer #1

(Remarks to the Author)

I would like to thank the authors for addressing my comments. Just some small, suggested edits.

- 1) In Figure 3, the use of brackets to refer to other sub-figures is not consistent and can lead to confusion
- 2) Thank you for the inclusion of additional information in the discussion. I still think that the discussion is rather short, but I will leave it on the editor to decide if this is an issue. I would expect to see a discussion of the method used in terms of its application in other systems and a brief description of the limitations of the approach and how likely they are to have affected the results of this investigation. I would also expect to see a deeper discussion on the currently mixed results, with evidence in humans not always supportive in terms of the use of lanosterol for cataract treatment. Finally, are the present results in a model organism directly translatable to humans? Based on the above, although I agree that the results obtained are promising, I suggest a more cautious final statement on how widely applicable that potential treatment is and the need for well conducted human clinical trials.
- 3) Thank you for the changes in previous Figure S8, now Figure S7. Has the change of using the mLNPs category changed the interpretation of the results? As you test a specific hypothesis, the difference of changes in the treated sample compared to the rest, instead of everything to everything, I suggest that you do not apply the Dunnett multiple comparison test. The one-way ANOVA is not the only test that can be used here and it only compares the 4 categories at the T+8 time-point, of what I understand. As the authors have data for other time points, if they can assume that the changes are linear, they can potentially compare the slopes from a linear regression of the 4 categories as well, using more of their collected data. This is just an idea and not a request.

Reviewer #2

(Remarks to the Author)

The revisions were not thorough and minimalistic.

Sample sizes are minimal for Fig 3 and Fig 4, many times only 2 biological replicates indicating experiments were not repeated.

In Fig 4a, the authors determine AC injection leads to minimal expression at 4 hours, and then maximum expression at 26 hours. However, the dose response data in 4f is from a 4-hour time point, and there is robust expression. These data are inconsistent. Probably as a result of low sample sizes.

When looking at the quantification for the rescue of the two rat models, which is in the supplemental instead of the main manuscript file, it's difficult to discern real rescue of the models. For instance in Fig S6, for untreated and eLNPs groups, cataract stage is averaging ~3, mLNPs reduce cataract stage to ~2....how meaningful is this? Additionally in Fig S7, mLNP treatment is not significantly different from Healthy, but it's also not significantly different from Untreated. It's difficult to discern the significant impact of the rescue.

Version 2:

Reviewer comments:

Reviewer #2

(Remarks to the Author)

The authors have now adequately addressed the revision comments. No further action is required.

We thank the reviewers for the constructive comments on our manuscript entitled "Ocular delivery of lipid nanoparticle-formulated mRNA encoding lanosterol synthase ameliorates cataract in rats". We have conducted additional experiments and included a point-by-point response to address these comments. New revisions highlighted in red in the revised manuscript have strengthened the quality of our manuscript.

REVIEWER COMMENTS

Reviewer #1 (Remarks to the Author):

The manuscript describes the use of a lipid nanoparticle carrying an mRNA encoding copy of human lanosterol synthase. The first part focuses on the production and optimisation of the delivery mechanism, while the second part focuses on its use in two mice models for cataract development. The work described seems carefully planned and executed, with relevant controls in all steps. As a geneticist, I cannot comment in depth on the development of the nanoparticle, but it appears appropriate and logical. The testing part is well planned and the results appear mostly convincing. We thank the reviewer for the positive feedback and comments. We have addressed each comment below.

I do not have any major comments for the work but a few extra checks and further explanation at specific points will help.

1. In Figure 3 the description of d seems to interrupt the description of c

We apologize for any confusion. While Figure 3c displays the quantified luminescence fluxes in the whole eyes from Figure 3a (whole-body imaging), Figure 3d shows those in the lenses and eyeball walls from Figure 3b (ex vivo imaging). To avoid confusion, we have updated this part in the revised manuscript (Page 8, line 162).

2. In Figure 5 d the mLNPs treated lens still seems to carry cataract changes which I am not sure agree with the statement that "the lens fiber cells in the healthy group exhibited a regular arrangement with no discernible gaps between cell nuclei (Figures 5d and 5e)". Could the author be a bit more descriptive by adding an arrow or other indication of what they comment on?

We thank the reviewer for the useful suggestion. Some arrows have been included in Figure 5d. In addition, we have revised related sentences to clarify the statement in the revised manuscript (Page 14, line 286).

3. The discussion is a bit short and does not comment on more recent findings looking at the use of lanosterol to restore lens clarity. Although, older work is cited such as Zhao et al (2015), More et al (2006) and Funk et al (2005), studies such as Shanmugam et al (2015), Daszynski (2019) and Hashimi et al (2024) are not included. Will it be possible for the authors to comment on how this study compares with these recent developments in their discussion and what that means for the use of lanosterol synthase as a treatment for cataract?

We have included a brief discussion on the more recent studies regarding the use of lanosterol to restore lens clarity in the revised manuscript (Page 16, lines 330 and 340).

4. In Figure S8, if the error bar represents the standard error of the mean, how is the test non-statistically significant with no error bar overlap? Also, should the authors test the mLNPs treated group (the focus of the work) against the other groups instead of comparing the healthy group to all others?

The error bar represents the standard error of the mean in Figure S8. In fact, there was no significant difference between the mLNPs-treated group and the healthy group when an unpaired, two-tailed Student's t-test or one-way analysis of variance (ANOVA) followed by a Tukey's test was used. In this section, the latter was adopted for multiple comparisons.

According to the reviewer's suggestion, we performed statistical analysis in the revised manuscript using one-way analysis of variance (ANOVA) followed by Dunnett's multiple comparison tests to compare the mLNPs-treated group to all others. Likewise, no significant difference between the mLNPs-treated group and the healthy group was observed. The exact P values have been provided in the revised Figure S7.

Reviewer #2 (Remarks to the Author):

It is estimated that over 95 million people suffer from cataracts globally. While cataract replacement surgery exists, many people don't have access to this surgery. Thus, novel therapeutics are still desired. The authors demonstrate that an LNP-mRNA therapeutic expressing human lanosterol

synthase can reduce cataracts in two rat models. This article is well-written. There is a good use of experimental controls. Data in Figure 5 with the sodium selenite-induced nuclear cataract model appears convincing. However, there are major concerns. Please see all the comments listed below. We thank the reviewer for the positive feedback and comments. We have addressed each comment below.

(Minor) Figure 1. It looks like a human eye is used in the diagram instead of a rodent eye. Please switch out to a rodent eye, with rodent sized lens.

We thank the reviewer for this suggestion to improve the diagram. Figure 1 has been redrawn and included in the revised manuscript.

Figure 2E. The authors claim pB-UC18 LNPs are mainly internalized into cells via clathrin-mediated endocytosis and micropinocytosis, but fail to discuss if this makes sense for lens internalization. Would cells in the lens use these mechanisms? Please add in the discussion.

We thank the reviewer for this comment and agree that it is critical to evaluate the possible mechanism for lens internalization. We conducted a similar experiment using SRA01/04 cells. The new result has been included in the revised manuscript.

Figure 2F&G. Data in F and G do not align. There is very little LSS protein observed in PBS treated cells (F), but in G there is a strong WB band. Which cells were used for the WB analysis? Also, authors could add the WB quantification to this figure instead of placing it in the supplemental figures.

We think that the green fluorescence intensity in Figure 2F is positively correlated with the exposure time during fluorescence imaging, and for WB, the band intensity in Figure 2G is positively correlated with the loading amount of total proteins.

The information about cells used for the WB analysis had been provided in the Experimental Section in the original manuscript, and now has been included in the legend of Figure 2G.

Figure S3 (western blot densitometry band quantification) has been removed from the supplemental file to the main text (Figure 2h).

(Major) Figure 4: Authors state: "Tissue immunofluorescence analysis indicated the treated lenses exhibited higher levels of expression of hLSS proteins compared to the untreated group after

formulation administration (Figure 4a).” The authors then go on to say, “The expression of hLSS proteins predominantly localized in the eyeball wall (such as corneal stroma, corneal endothelium, and choroid), the lens epithelium, and lens cortex rather than the nuclear region of the lens 24 h after a single AC injection of mLNPs (Figure 4a).”

I don't see data that supports these statements. I don't see any expression in the lens epithelium, very little in the cortex, and why would there be any expression in the choroid after an AC injection? Also, the authors are expressing human LSS, is that different than rat LSS? Does the antibody detect both rat and human versions? Would it be helpful to distinguish between endogenous levels and the overexpressed levels from the LNPs? The authors should clarify. Overall, there is very little evidence that the LNPs transfect and express mRNA in the lens. The most convincing data is the luciferase data, but luciferase data often results in false positives and must be supported by immunofluorescence or WB data. It did look like there was expression in the cornea. Could that then lead to increased HPLC levels? Immunofluorescence data needs to be more convincing to claim that the LNPs transfect the lens.

We thank the reviewer for pointing out these questions. By tuning the fluorescence signal to the same extent for both the unjected group and the mLNPs-treated group, we found successful hLSS expression in the lens epithelium and lens cortex of the mLNPs-treated rats after an AC injection (Representative fluorescence signals in Figure 4a were indicated by white triangles). Regarding hLSS expression in the choroid, we speculate that mLNPs enter the choroid through the uveoscleral pathway after an AC injection as the choroid has the highest blood flow per gram of any tissue.

Although human LSS is different from rat LSS, both are highly conserved and can be detected by the antibody used in this study according to the manufacturers' instructions. We agree that it is helpful to distinguish between endogenous levels and the overexpressed levels from the LNPs. However, as shown in Figure 4, endogenous rat LSS from the ocular tissues (except corneal epithelium) of unjected rats is too low to be detected by this antibody, and the overexpressed human LSS encoded by exogenous hLSS mRNA can be detected to distinguish endogenous rat LSS. Together, hLSS expression can be found in corneal stroma, corneal endothelium, lens

epithelium, lens cortex, and choroid (revised Figure 4), all of which may contribute to the increased HPLC levels.

Figure 5: These data are convincing. What are your statistics for these data, how many animals, how many replicates? In the discussion, please discuss how effective this treatment is compared to others. How translatable is this technology? How different is this model in comparison to the ways most patients get cataracts (genetics, age, trauma, etc)?

We performed statistical analysis using two-way analysis of variance (ANOVA) followed by a Sidak's test to compare the injected eye to the uninjected eye of the same rat. n= 3 rats for the healthy group; n= 6 rats for the untreated group; n= 3 rats for the eLNPs-treated group; n= 8 rats for the mLNPs-treated group. A detailed description of this technology has been included in the Conclusion section (Page 16, lines 330 and 340).

Figure 6: This model is less convincing and so are improved outcomes.

D-galactose-induced cataract rat model has been widely used to mimic diabetic cataracts, and the outcomes in this study were similar to or better than those reported by previous studies¹⁻⁵.

References

- [1] Zhong L, Wang T, Wang T, Cheng H, Deng J, Ye H, Li W, Ling S. Characterization of an i.p. D-galactose-induced cataract model in rats. *J Pharmacol Toxicol Methods*. 2021, **107**:106891.
- [2] Velpandian T, Gupta P, Ravi AK, Sharma HP, Biswas NR. Evaluation of pharmacological activities and assessment of intraocular penetration of an ayurvedic polyherbal eye drop (Itone™) in experimental models. *BMC Complement Altern Med*. 2013, **13**:1.
- [3] Sun J, Wang B, Hao Y, Yang X. Effects of calcium dobesilate on Nrf2, Keap1 and HO-1 in the lenses of D-galactose-induced cataracts in rats. *Exp Ther Med*. 2018, **15**:719-722.
- [4] Nagaya M, Yamaoka R, Kanada F, Sawa T, Takashima M, Takamura Y, Inatani M, Oki M. Histone acetyltransferase inhibition reverses opacity in rat galactose-induced cataract. *PLoS One*. 2022, **17**:e0273868.

[5] Wang Y, Tseng Y, Chen K, Wang X, Mao Z, Li X. Reduction in Lens Epithelial Cell Senescence Burden through Dasatinib Plus Quercetin or Rapamycin Alleviates D-Galactose-Induced Cataract Progression. *J Funct Biomater.* 2022, **14**:6.

(Minor) Figure S3: Which figure is this dataset referring to? Please clarify.

The dataset in Figure S3 (western blot densitometry band quantification) referring to Figure 2g has been removed from the supplemental file to the main text (Figure 2h).

Figure S7: Is there any significant difference between the orange bars between groups? It looks like you are getting a significant difference in cataract stage in mLNPs because injection – group has a worse cataract stage (increased compared to untreated and eLNPs). What are the sample sizes?

We apologize for any confusion. Given the high rat-to-rat variability in the selenite-induced cataract rat model, one eye was treated with eLNPs or mLNPs while the other eye from the same rat was left untreated for comparison to eliminate individual differences (such description had been included in the original manuscript). Therefore, no statistical analysis was performed between the orange bars. The situation where the uninjected eye from the mLNPs-treated group has a worse cataract stage than the uninjected eye from the untreated and eLNPs-treated group just right further demonstrates superior efficacy of mLNPs even at a worse cataract stage.

The sample sizes in Figure S7 (now revised Figure S6) are the same as those in Figure 5.

We thank the reviewers for the constructive comments. We have included a point-by-point response to address these comments. New revisions marked in red in the revised manuscript have further strengthened the quality of our manuscript.

REVIEWER COMMENTS

Reviewer #1 (Remarks to the Author):

I would like to thank the authors for addressing my comments. Just some small, suggested edits.

1) In Figure 3, the use of brackets to refer to other sub-figures is not consistent and can lead to confusion.

We thank the reviewer for this suggestion to improve the quality of Figure 3b. To avoid confusion and highlight the predominant FLuc expression in the lenses, we have revised this figure.

2) Thank you for the inclusion of additional information in the discussion. I still think that the discussion is rather short, but I will leave it on the editor to decide if this is an issue. I would expect to see a discussion of the method used in terms of its application in other systems and a brief description of the limitations of the approach and how likely they are to have affected the results of this investigation. I would also expect to see a deeper discussion on the currently mixed results, with evidence in humans not always supportive in terms of the use of lanosterol for cataract treatment. Finally, are the present results in a model organism directly translatable to humans? Based on the above, although I agree that the results obtained are promising, I suggest a more cautious final statement on how widely applicable that potential treatment is and the need for well conducted human clinical trials.

An extensive discussion on these themes have been included in the Discussion section, according to the reviewer's suggestion.

3) Thank you for the changes in previous Figure S8, now Figure S7. Has the change of using the mLNPs category changed the interpretation of the results? As you test a specific hypothesis, the difference of changes in the treated sample compared to the rest, instead of everything to everything, I suggest that you do not apply the Dunnett multiple comparison test. The one-way ANOVA is not the only test that can be used here and it only compares the 4 categories at the T+8 time-point, of what I understand. As the authors have data for other time points, if they can assume that the changes are linear, they can potentially compare the slopes from a linear regression of the

4 categories as well, using more of their collected data. This is just an idea and not a request.

We thank the reviewer for the helpful suggestion. We have acquired the slopes from a linear regression of the 4 categories according to the reviewer's suggestion and found that the turbidity area including annular equatorial-like vesicles and radial turbidity within the lenses within the lenses rapidly diminished over time following administration of mLNPs, as evidenced by the slit-lamp photographs and the slopes of linear regression (Figs. 6b and 6c).

Reviewer #2 (Remarks to the Author):

The revisions were not thorough and minimalistic.

Sample sizes are minimal for Fig 3 and Fig 4, many times only 2 biological replicates indicating experiments were not repeated.

We thank the reviewer for pointing out this question. It is a common practice to use two rodents (or even one rodent) in the mRNA field to improve the welfare of laboratory animals [1-7], which implicates to the second R, Reduction, of the 3Rs Principles of Replacement, Reduction, and Refinement in animal research (Ref 1, $n = 1$ for Figures 2c and 4c; Ref 2, $n = 2$ for Figures 2d-2i; Ref 3, $n = 2$ for Figures 1f and 1g; Ref 4, $n = 2$ for Figures 2h; Ref 5, $n = 2$ for Figures F4a-4d; Ref 6, $n = 2$ for Figures 3d and 3e; Ref 7, $n = 2$ for Figures 6b).

In addition, this animal experiment involves burdensome ocular micromanipulation under anesthesia for four different routes of administration (each route of administration is examined at three specific time points). To minimize the number of rats used in this study and the variability induced by burdensome ocular micromanipulation while still obtain enough information from each rat, we used four rats for each route of administration at the beginning of this animal experiment. Following whole-body bioluminescence imaging (4 h), 2 out of 4 rats from each group were sacrificed at 4 h for ex vivo imaging, and the remaining 2 rats were subjected to whole-body bioluminescence imaging repeatedly over time (26 h and 48 h) and sacrificed at 48 h for ex vivo imaging. These details now have now been included in the legend of Figures 3a and 3b.

Judging from the error bars in Figure 3c, we estimate that the variation among values collected from four routes of administration is relatively small, which should reflect the delivery performance of formulations. Additionally, we also have clarified the fact about consistency below, which is irrelevant to sample sizes.

In Fig 4a, the authors determine AC injection leads to minimal expression at 4 hours, and then maximum expression at 26 hours. However, the dose response data in 4f is from a 4-hour time point, and there is robust expression. These data are inconsistent. Probably as a result of low sample sizes.

We apologize for any confusion. From quantitative analysis showed in both Fig 3c and Fig S3 (now Figure 3f), we can conclude that the data in Fig 3a (not Fig 4a) and Fig 3f (not Fig 4f, now Figure

3e) are not contradictory to each other. It is the different colour gradients in both figures ($5 \times 10^5 \sim 2 \times 10^6$ vs $1 \times 10^4 \sim 1 \times 10^6$) that create an optical illusion. To provide a convenient way for the direct comparison, Figure S3 has now been removed from the supplemental file to the main text (now Figure 3f).

When looking at the quantification for the rescue of the two rat models, which is in the supplemental instead of the main manuscript file, it's difficult to discern real rescue of the models. For instance in Fig S6, for untreated and eLNPs groups, cataract stage is averaging ~ 3 , mLNPs reduce cataract stage to ~ 2how meaningful is this? Additionally in Fig S7, mLNP treatment is not significantly different from Healthy, but it's also not significantly different from Untreated. It's difficult to discern the significant impact of the rescue.

We apologize for any inconvenience. Both Figures S6 and S7 have now been removed from the supplemental file to the main text (now Figures 5d and 6c, respectively).

We also apologize for any confusion. For Fig S6 (now Figures 5d), it should be noted that rats were randomly divided into four groups before the phenotype of cataract occurring. Thus, we use two eyes of a rat (one data point in the blue bar and the other in the orange bar for each group) rather than one eye of each individual rat for intragroup comparison to eliminate the high rat-to-rat intergroup variability (evidenced by all data point in the blue or orange bar) in the selenite-induced cataract rat model. Such description had been included in the original manuscript. The two blue bars from the untreated group in Figures 5d are just used to tell readers that sodium selenite induces similar severity of nuclear cataract in both eyes of the same rat, and the two bars from the eLNPs group in Figures 5d demonstrates that eLNPs are ineffective. In addition, it is worth emphasizing that small decreases in cataract stage (e.g. from 3 to 2) represent significant improvements in cataract development for the sodium selenite-induced nuclear cataract model. Collectively, any intergroup bars are not suitable for comparison. The worse cataract stage evidenced by the uninjected eye (injection -) from the mLNPs group just right further demonstrates the superior efficacy of mLNPs (injection +).

For Fig S7 (now Figures 6c), we have acquired the slopes from a linear regression of the 4 categories according to the reviewer #1's suggestion and found that the turbidity area including annular equatorial-like vesicles and radial turbidity within the lenses within the lenses rapidly

diminished over time following administration of mLNPs, as evidenced by the slit-lamp photographs and the slopes of linear regression (Figs. 6b and 6c).

References

- [1] He Z, Le Z, Shi Y, Liu L, Liu Z, Chen Y. A Multidimensional Approach to Modulating Ionizable Lipids for High-Performing and Organ-Selective mRNA Delivery. *Angew Chem Int Ed Engl.* 2023; 62(43):e202310401.
- [2] Li B, Raji IO, Gordon AGR, Sun L, Raimondo TM, Oladimeji FA, Jiang AY, Varley A, Langer RS, Anderson DG. Accelerating ionizable lipid discovery for mRNA delivery using machine learning and combinatorial chemistry. *Nat Mater.* 2024; 23(7):1002-1008.
- [3] Li B, Manan RS, Liang SQ, Gordon A, Jiang A, Varley A, Gao G, Langer R, Xue W, Anderson D. Combinatorial design of nanoparticles for pulmonary mRNA delivery and genome editing. *Nat Biotechnol.* 2023; 41(10):1410-1415.
- [4] Han X, Alameh MG, Gong N, Xue L, Ghattas M, Bojja G, Xu J, Zhao G, Warzecha CC, Padilla MS, El-Mayta R, Dwivedi G, Xu Y, Vaughan AE, Wilson JM, Weissman D, Mitchell MJ. Fast and facile synthesis of amidine-incorporated degradable lipids for versatile mRNA delivery in vivo. *Nat Chem.* 2024; 16(10):1687-1697.
- [5] Liu S, Wang X, Yu X, Cheng Q, Johnson LT, Chatterjee S, Zhang D, Lee SM, Sun Y, Lin TC, Liu JL, Siegwart DJ. Zwitterionic Phospholipidation of Cationic Polymers Facilitates Systemic mRNA Delivery to Spleen and Lymph Nodes. *J Am Chem Soc.* 2021; 143(50):21321-21330.
- [6] Eygeris Y, Gupta M, Kim J, Jozic A, Gautam M, Renner J, Nelson D, Bloom E, Tuttle A, Stoddard J, Reynaga R, Neuringer M, Lauer AK, Ryals RC, Sahay G. Thiophene-based lipids for mRNA delivery to pulmonary and retinal tissues. *Proc Natl Acad Sci U S A.* 2024; 121(11):e2307813120.
- [7] Li Z, Amaya L, Pi R, Wang SK, Ranjan A, Waymouth RM, Blish CA, Chang HY, Wender PA. Charge-altering releasable transporters enhance mRNA delivery in vitro and exhibit in vivo tropism. *Nat Commun.* 2023; 14(1):6983.